# STM4D: 4D Occupancy Forecasting with 2D and 3D Spatio-Temporal Modeling

## Abstract

Vision-based 4D occupancy forecasting enables autonomous vehicles to predict future 3D semantic scenes from historical multi-view images, which is critical for driving safety. While current methods show promising results, the potential of simultaneous 2D and 3D spatio-temporal modeling and leveraging temporal cues from 2D multi-view image sequences to improve 4D occupancy prediction remains unexplored, presenting a critical bottleneck for advancing performance. To address this gap, we introduce STM4D, a novel framework for 4D occupancy prediction that jointly models temporal dynamics in both voxel-based representations and multi-view image sequences, while explicitly incorporating feature interaction between the two complementary branches. Our framework incorporates three core components: 1) A 3D Spatio-Temporal (3DST) module that learns volumetric dynamics from historical voxel states to predict future voxel states; 2) A 2D Spatio-Temporal (2DST) module employing an auxiliary segmentation forecasting task to enhance temporal semantic consistency; 3) A Spatio-Temporal Interaction Modeling (STIM) module that enables camera-agnostic feature interaction between 2D and 3D representations. The unified architecture is trained end-to-end and establishes new state-of-the-art performance on both Occ3D-nuScenes and Cam4DOcc benchmarks.

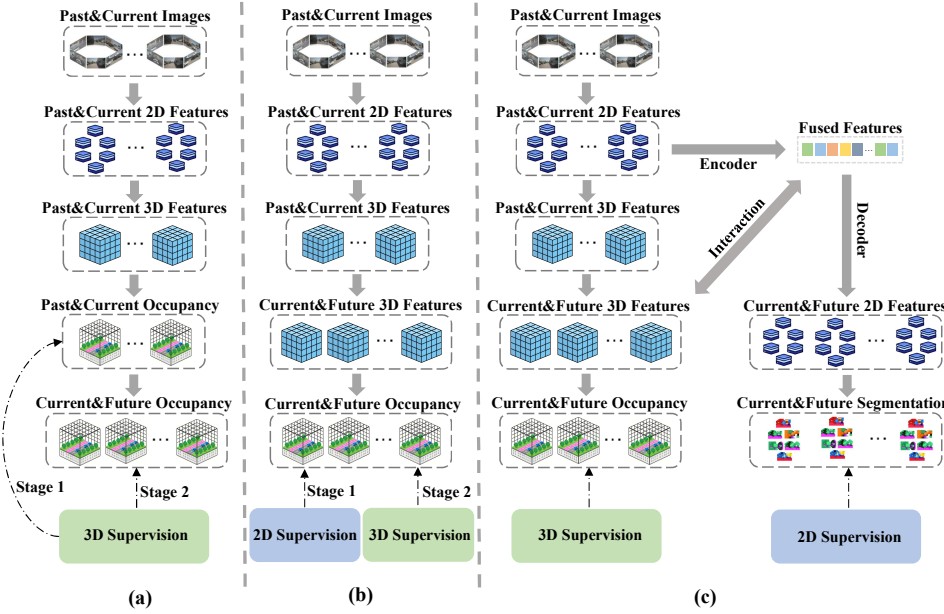

Figure 1: The comparison of the proposed method with other frameworks: (a) 3D occupancy-based forecasting model; (b) 3D feature-based forecasting model with 3D and 2D labels as supervision; (c) Our proposed framework adopts both 2D and 3D spatio-temporal modeling processes with extra 2D-3D interactions.

# 1 INTRODUCTION

Vision-centric 3D occupancy prediction estimates the 3D semantics of each voxel in the entire scene from input multi-view images, which plays a crucial role in autonomous driving (Tong et al., 2023; Ma et al., 2024b; Zhang et al., 2023b; Tian et al., 2023; Huang et al., 2023). For a better understanding of the dynamic world and providing clues of the future 3D scenes, vision-centric 4D occupancy forecasting (Ma et al., 2024a) is further introduced to predict the future 3D occupancy given previous observations. It helps to understand the world dynamics and advance the downstream tasks, such as collision avoidance and route planning (Min et al., 2024).

Existing 4D occupancy forecasting methods mainly adopt an occupancy-based forecasting framework (Yan et al., 2024) (see Figure 1(a)) : First, obtain the past and current occupancy with a pre-trained 3D occupancy model; then, feed them into a forecasting model to predict the future occupancy, which first encodes the occupancy into tokens and generates future tokens with an autoregressive model, and finally decodes future tokens into future occupancies. However, such iterative encoding and decoding operations can lead to progressive information loss. Moreover, this approach heavily relies on 3D annotations and cannot be trained in an end-to-end manner. The recently proposed PreWorld (Li et al., 2025) is a semi-supervised model for 4D occupancy forecasting. It introduces a straightforward state-conditioned forecasting module and leverages 2D supervision through voxel rendering (see Figure 1(b)). Although PreWorld reduces the dependency on densely annotated 3D occupancy labels, it fails to effectively utilize the temporal information inherent in 2D image sequences for 4D occupancy forecasting due to its reliance on a purely supervision-driven approach. Overall, existing approaches exhibit a notable deficiency in exploring integrated 2D and 3D spatio-temporal modeling frameworks and overlook the essential interactions between these two paradigms to better leverage temporal cues from the 2D branch for more effective spatio-temporal modeling, which fundamentally constrains their long-term forecasting capabilities.

We propose **STM4D**, a unified architecture for joint 2D-3D spatio-temporal modeling that explicitly bridges complementary representations through synchronized cross-modal interaction. Our framework introduces dedicated interaction mechanisms between temporally aligned 2D and 3D branches (Figure 1(c)), establishing a new paradigm for holistic 4D scene understanding that simultaneously captures semantic dynamics in image space and geometric evolution in volumetric representations. The complete architecture is detailed in Figure 2.

Specifically, we propose a 3D Spatio-Temporal (3DST) module to capture spatio-temporal dependencies in volumetric representations. This module predicts future voxel states based on historical voxel states derived from multi-view images, enhancing the modeling of 3D scene dynamics. Subsequently, we introduce a 2D Spatio-Temporal (2DST) module, which incorporates an auxiliary task aimed at predicting future multi-view image segmentation sequences from historical image features. The semantic segmentation labels are automatically annotated using Segment Anything (SAM) (Kirillov et al., 2023). Finally, we present a Spatio-Temporal Interaction Modeling (STIM) module to facilitate cross-modal feature interaction between 2D and 3D representations without relying on camera parameters, further improving the performance of 4D occupancy forecasting. The entire framework is trained in an end-to-end manner, eliminating the need for staged optimization.

The contributions of the paper are summarized as follows:

- We propose a novel 3D spatio-temporal modeling (3DST) module that effectively learns volumetric dynamics from historical voxel states to predict future states, thereby advancing 4D occupancy forecasting capabilities.
- We propose a 2D Semantic Spatio-Temporal (2DST) module that predicts future semantic maps while facilitating semantically-aware temporal regularization learning in 2D space.
- Instead of adopting the 3D and 2D spatio-temporal modeling branches separately, we introduce a Spatio-Temporal Interaction Module (STIM) that enables camera-agnostic 2D-3D interactions through cross-modal feature alignment, thereby enabling more direct and effective utilization of spatio-temporal cues from the 2D branch.
- Extensive experiments demonstrate that the proposed STM4D method achieves state-of-the-art performance on both the Occ3D-nuScenes and Cam4DOcc benchmarks for 4D occupancy forecasting. Furthermore, the design enables the model to maintain strong performance even with limited 3D annotations.

## 2 RELATED WORK

**3D Occupancy Prediction.** Vision-based 3D occupancy prediction Tong et al. (2023); Ma et al. (2024b); Tang et al. (2024); Zhao et al. (2024); Ouyang et al. (2024); Zhang et al. (2025); Zhu et al. (2024); Lu et al. (2024) has attracted considerable attention in autonomous driving for its capability to generate dense 3D environmental representations that surpass traditional bounding box methods. This approach estimates semantic occupancy states from multi-view camera images, enabling more comprehensive scene understanding crucial for safe navigation. To address the challenges of acquiring dense 3D annotations, several methods have explored using 2D information as alternative supervision signals. Specifically, Vampire (Xu et al., 2024) regulates intermediate 3D volume features by incorporating rendered camera-view depth and semantic information during training. Extending this direction, RenderOcc (Pan et al., 2024a) and SelfOcc (Huang et al., 2024) employ more radical strategies that depend exclusively on 2D supervision, completely eliminating the need for 3D labels to reduce annotation costs and complexity. However, these methods utilize 2D information merely from a supervisory perspective, which fails to fully exploit its potential. *In contrast, our approach focuses on 2D and 3D spatiotemporal modeling and the interaction between 2D and 3D branches, enabling more direct and effective utilization of 2D information.*

**4D Occupancy Forecasting.** Compared to the 3D case, 4D occupancy forecasting places greater emphasis on spatio-temporal information modeling, as it requires the model to anticipate how the scene will evolve. Cam4DOcc (Ma et al., 2024a) proposed a strong baseline and a comprehensive benchmark for 4D occupancy forecasting. DriveWorld (Min et al., 2024) provided a pre-training framework considering both spatial and temporal information for 4D downstream tasks in a driving scenario. Drive-OccWorld (Yang et al., 2025) proposed an efficient semantic and motion conditional normalization to enhance the historical BEV feature and conducted temporal cross-attention on it for better temporal modeling. OccWorld (Zheng et al., 2025) decomposed the scene into world tokens and proposed a spatial-temporal generative transformer to aggregate multi-scale token embeddings. OccProphet (Chen et al., 2025) proposed a novel tripling operation to decompose the voxel features into scene, height, and BEV components separately, enabling lightweight spatio-temporal feature interaction and significantly reducing the computational cost. EfficientOCF (Xu et al.) employs task-level spatiotemporal decoupling to enhance 4D occupancy, but it relies on additional 3D annotations. The above methods have explored temporal modeling in the 3D domain for 4D Occupancy, yet none of them have investigated leveraging 2D labels to enhance performance or reduce reliance on 3D annotations. Then, PreWorld (Li et al., 2025) proposed a semi-supervised 4D occupancy forecasting pipeline that introduces 2D rendering supervision for predicted future 3D volume features and fine-tunes the network using limited 3D occupancy labels, thereby reducing the dependency on dense 3D annotations. However, merely utilizing 2D information from a supervisory perspective fails to directly leverage the inherent temporal cues present in 2D sequences. *To address this gap, we introduce a novel framework that jointly models temporal dynamics in both voxel-based representations and multi-view image sequences, while explicitly incorporating feature interaction between the two complementary branches to better leverage temporal cues in 2D sequences.*

**Spatio-Temporal Modeling in Video Prediction.** Spatio-Temporal Modeling has been well studied in the 2D video prediction tasks Guen & Thome (2020); Wang et al. (2020); Yu et al. (2022); Girdhar & Grauman (2021); Liu et al. (2023); Benaim et al. (2020). Many works try to solve it from different perspectives, such as architecture or semantics. MAU (Chang et al., 2021) designed an efficient attention and fusion module to aggregate temporal information at different levels in order to have a broader temporal receptive field. MotionRNN (Wu et al., 2021) proposed a novel MotionGRU unit that decomposes object motion into transient variations and motion trends, and integrates it into an RNN framework to achieve more accurate motion prediction. SADM (Bei et al., 2021) observed that different semantic regions have different dynamic characteristics. Based on this observation, they proposed a newly designed semantic-aware dynamic model that predicts complete video frames with both semantic and geometric consistency. PastNet (Wu et al., 2024) pointed out that prior methods often overlook the physical priors in videos. To address this, they introduced spectral convolution in the Fourier domain to embed inductive biases from physical laws and designed an efficient discrete spatio-temporal (DST) module to explore spatio-temporal information. DFDNet (Gan et al., 2025) first filters out transient high-dynamic information, which may contain irrelevant noise, and then extracts temporal dependencies from the filtered sequences. *In our work, we explore enhancing 4D occupancy forecasting by leveraging 2D temporal information through interactive modeling between 3D voxel sequences and camera view video sequences.*

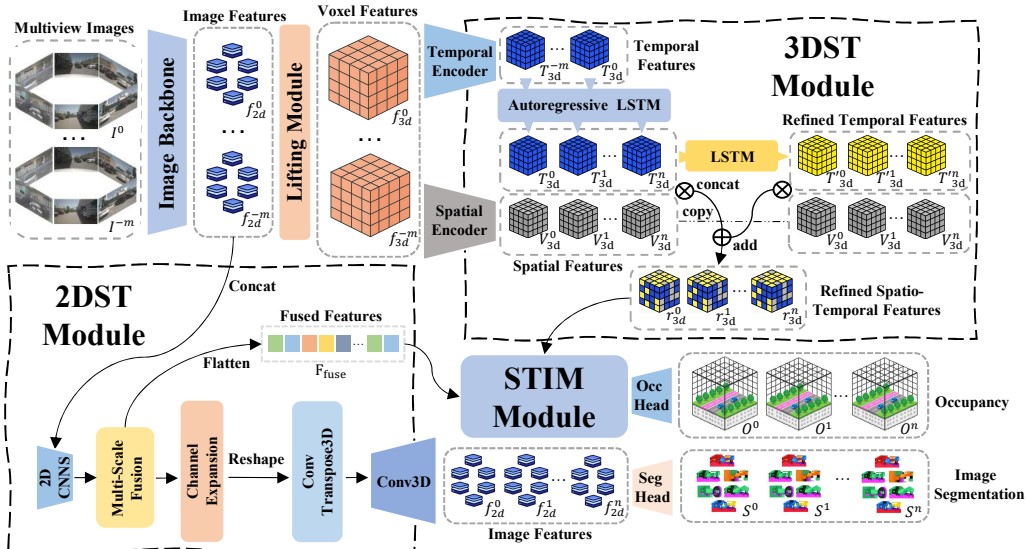

Figure 2: The overall pipeline of our STM4D: First, we extract multi-camera image features from the past and current frames. The bottom 2D branch processes image features through the 2DST module and utilizes a segmentation head to predict future semantic segmentation maps. The top 3D branch projects image features into 3D space and refines them via the 3DST Module, generating future refined spatio-temporal features. Both the fused features from 2DST and refined spatio-temporal features from 3DST then interact through the STIM Module and are finally passed to the occupancy head for current and future occupancy prediction.

# 3 4D OCCUPANCY FORECASTING WITH SPATIO-TEMPORAL MODELING

## 3.1 MODEL ARCHITECTURE

As illustrated in Figure 2, our proposed STM4D model takes past and current multi-view images $\{I^i\}_{i=-m}^0$ as input, where the superscript $i$ denotes the timestamp (with $i = 0$ corresponding to the current frame). Throughout this paper, superscripts on all symbols represent timestamps, with both $m > 0$ and $n > 0$ denoting the lengths of historical and future sequences, respectively. First, an image backbone (e.g., ResNet (He et al., 2016)) extracts image features $\{f_{2d}^i\}_{i=-m}^0$ for each timestamp. These features are then fed into both the 2D branch and the 3D branch, respectively.

In the 3D branch, we use LSS (Philion & Fidler, 2020) to lift the 2D image features $\{f_{2d}^i\}_{i=-m}^0$ into 3D space, obtaining a set of voxel features $\{f_{3d}^i\}_{i=-m}^0$. These voxel features $\{f_{3d}^i\}_{i=-m}^0$ are then processed by a novel **3D Spatio-Temporal Modeling (3DST)** module to generate refined spatio-temporal features $\{r_{3d}^i\}_{i=0}^n$ for future time steps.

In the 2D branch, the image features $\{f_{2d}^i\}_{i=-m}^0$ are passed through a **2D Spatio-Temporal Modeling (2DST)** module for temporal enhancement and future prediction. The output features $\{f_{2d}^i\}_{i=0}^n$ are then processed by a segmentation head to produce multi-view semantic segmentation maps $\{S^i\}_{i=0}^n$. Meanwhile, multi-scale fused features are flattened across both view and spatial dimensions to form the fused features $F_{\text{fuse}}$, which are subsequently passed to the Spatio-Temporal Interaction Module (STIM).

Finally, our proposed **Spatio-Temporal Interaction Module (STIM)** integrates the fused features $F_{\text{fuse}}$ from the 2DST module and the refined spatio-temporal features $\{r_{3d}^i\}_{i=0}^n$ from the 3DST module through cross-domain interaction. The resulting representation is forwarded to the occupancy head to jointly predict current and future occupancy states $\{O^i\}_{i=0}^n$.

## 3.2 3D SPATIO-TEMPORAL MODELING

We propose a new **3D Spatio-Temporal (3DST) module** that learns volumetric dynamics from historical voxel states to predict future voxel states, thereby advancing 4D occupancy forecasting

capabilities (see Figure 2). The input to our 3DST consists of historical and current Voxel Features $\{f_{3d}^i\}_{i=-m}^0$ obtained from the Lifting Module.

To jointly capture motion patterns and spatial structures, we process the voxel features through parallel temporal and spatial encoding pathways. The temporal encoder models dynamic changes across frames, while the spatial encoder integrates cross-frame information through channel-wise concatenation:

$$\{T_{3d}^i\}_{i=-m}^0 = \mathcal{G}_t(\{f_{3d}^i\}_{i=-m}^0), \tag{1}$$

$$\{V_{3d}^i\}_{i=0}^n = \mathcal{G}_s\left(\text{Concat}(\{f_{3d}^i\}_{i=-m}^0)\right), \tag{2}$$

where $\mathcal{G}_t$ and $\mathcal{G}_s$ denote 3D CNN-based temporal and spatial encoders respectively (e.g., ResNet3D for both modules).

The temporal features $\{T_{3d}^i\}_{i=-m}^0$ undergo sequential processing through autoregressive prediction and refinement stages: the $\text{LSTM}_{\text{AR}}$ module forecasts future temporal features, while the $\text{LSTM}_{\text{Refine}}$ layer enhances their temporal consistency:

$$\{T_{3d}^i\}_{i=0}^n = \text{LSTM}_{\text{AR}}(\{T_{3d}^i\}_{i=-m}^0), \tag{3}$$

$$\{T_{3d}'^i\}_{i=0}^n = \text{LSTM}_{\text{Refine}}(\{T_{3d}^i\}_{i=0}^n). \tag{4}$$

For each timestamp, we concatenate the spatial features $V_{3d}^i$ with both temporal features $T_{3d}^i$ and refined temporal features $T_{3d}'^i$, followed by a summation to obtain refined spatio-temporal features:

$$r_{3d}^i = \text{Sum}\Big(\text{Concat}(V_{3d}^i, T_{3d}^i),\ \text{Concat}(V_{3d}^i, T_{3d}'^i)\Big), \quad i = 0, 1, ..., n. \tag{5}$$

It is noteworthy that our LSTM architecture follows 3D ConvLSTM (Shi et al., 2015), with further details provided in the appendix A.1. Through this design, the temporal encoder and autoregressive LSTM capture dynamic evolution, while the spatial encoder preserves cross-frame structure. The fusion step ensures that spatial cues compensate for the loss of spatial information during temporal modeling, leading to robust 4D voxel forecasting.

### 3.3 2D SPATIO-TEMPORAL MODELING

Although some approaches, such as PreWorld (Li et al., 2025), attempt to leverage 2D supervision by rendering occupancy into semantic maps or depth images, they overlook the temporal cues presented in 2D sequences. To address this issue, we propose a **2D Spatio-Temporal (2DST) module** (see Figure 2), which introduces an auxiliary task that predicts future multi-view semantic segmentation maps $\{S^i\}_{i=0}^n$ from historical multi-view image features $\{f_{2d}^i\}_{i=-m}^0$. This design enables the model to learn semantically-aware temporal regularization in 2D space.

We first take the historical image features $\{f_{2d}^i\}_{i=-m}^0$ as input. These features are concatenated along the feature channel dimension and passed through a DenseNet-based 2D CNN $\mathcal{D}$ to obtain multi-scale representations that comprehensively unify spatio-temporal information from historical frames:

$$F_{\text{ms}} = \mathcal{D}\big(\text{concat}(\{f_{2d}^i\}_{i=-m}^0)\big). \tag{6}$$

The resulting multi-scale features are processed by a multi-scale fusion module $\mathcal{M}$, where each scale is aligned to the smallest resolution, concatenated, and fused. The flattened output is defined as the fused feature representation:

$$F_{\text{fuse}} = \text{Flatten}(\mathcal{M}(F_{\text{ms}})). \tag{7}$$

This fused feature is later forwarded to the Spatio-Temporal Interaction Module (STIM) introduced in Section 3.4.

In parallel, another branch expands channels and reshapes $F_{\text{ms}}$ to produce future multi-frame representations. These representations are then progressively upsampled by 3D transposed convolutions and refined with 3D convolutions, yielding predicted future 2D features $\{f_{2d}^i\}_{i=0}^n$. This process of 3D convolution-based upsampling and refinement explicitly models spatio-temporal relationships across multiple frames.

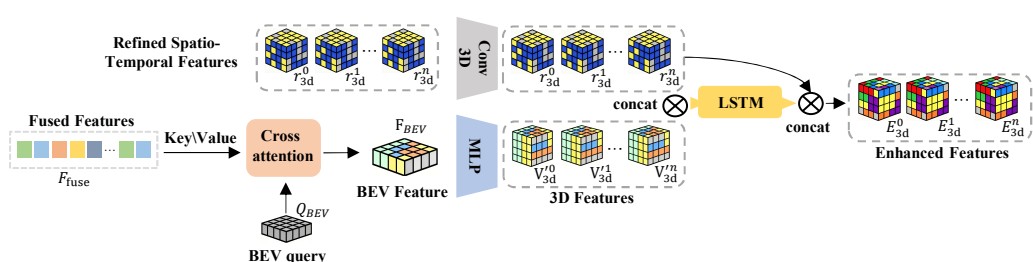

Figure 3: Our STIM module leverages refined spatio-temporal features from 3DST and fused features from 2DST as inputs, establishes spatio-temporal interactions between 2D and 3D branches without relying on any camera parameters, and ultimately outputs enhanced features.

### 3.4 SPATIO-TEMPORAL INTERACTION MODELING

We further propose a novel Spatio-Temporal Interaction Module (STIM), enabling explicit cross-dimensional interaction to directly leverage temporal cues derived from 2D spatio-temporal modeling. The STIM module explicitly captures interdependencies between the fused features $F_{\text{fuse}}$ from the 2DST module and the refined spatio-temporal features $r_{3d}^i$ generated by the 3DST module, establishing a bidirectional bridge between these complementary representations (see Figure 3).

Specifically, the fused features $F_{\text{fuse}}$ serve as an intermediate representation in 2DST that bridges historical information and future predictions, encapsulating spatio-temporal cues from the 2D branch. These features are utilized as both key and value inputs to the cross-attention module, while a learnable bird's-eye-view (BEV) query $Q_{\text{BEV}}$ is fed as the query input. Through this process, structured BEV features $F_{\text{BEV}}$ are generated:

$$F_{\text{BEV}} = \text{CrossAttn}(Q_{\text{BEV}}, F_{\text{fuse}}). \tag{8}$$

It is worth noting that we did not use any camera parameters throughout this process. Since the fused features $F_{\text{fuse}}$ have a small size, we abandoned projection-based local interaction and adopted global interaction. Specifically, each query $Q_{\text{BEV}}$ interacts with the entire fused features $F_{\text{fuse}}$ through attention mechanisms, thereby eliminating the need for any camera parameters. Subsequently, a lightweight MLP is employed to predict the feature distribution along the height dimension from BEV features $F_{\text{BEV}}$. This height-aware feature representation is then replicated across the temporal dimension to generate the 3D features $\{V_{3d}^{\prime i}\}_{i=0}^n$.

Subsequently, the refined spatio-temporal features $\{r_{3d}^i\}_{i=0}^n$ are processed by a 3D convolution to reduce their channel dimensionality to get $\{\tilde{r}_{3d}^i\}_{i=0}^n$. These features $\{\tilde{r}_{3d}^i\}_{i=0}^n$ are then concatenated with the 3D features $\{V_{3d}^{\prime i}\}_{i=0}^n$ and fed into an LSTM module. The output of the LSTM is further concatenated with the channel-reduced refined spatio-temporal features $\{\tilde{r}_{3d}^i\}_{i=0}^n$ to form the enhanced features $\{E_{3d}^i\}_{i=0}^n$.

$$\{E_{3d}^i\}_{i=0}^n = \text{Concat}\left(\text{LSTM}\left(\text{Concat}\left(\{\tilde{r}_{3d}^i\}_{i=0}^n, \{V_{3d}^{\prime i}\}_{i=0}^n\right)\right), \{\tilde{r}_{3d}^i\}_{i=0}^n\right). \tag{9}$$

These enhanced features are subsequently fed into an occupancy head to predict the final occupancy output $\{O^i\}_{i=0}^n$.

## 4 EXPERIMENTS AND RESULTS

### 4.1 DATASETS AND METRICS

**Occ3D-nuScenes** (Tian et al., 2023) extends nuScenes (Caesar et al., 2020) with voxel annotations in $[-40, 40]\text{m} \times [-40, 40]\text{m} \times [-1, 5.4]\text{m}$ (ego-centric), discretized to $200 \times 200 \times 16$ voxels at 0.4m resolution. Following standard protocols, we use mIoU (primary) and IoU metrics for evaluation.

**Cam4DOcc** (Ma et al., 2024a) utilizes 700 training scenes (23,930 sequences) and 150 test scenes (5,119 sequences) from nuScenes/nuScenes-Occupancy (Wang et al., 2023a). The 7-frame config-uration (3 observed + 4 predicted) covers $[-51.2, 51.2]\text{m} \times [-51.2, 51.2]\text{m} \times [-5, 3]\text{m}$ with 0.2m

Table 1: Results comparison on mIoU and IoU at 1s, 2s, and 3s time horizons on Occ3D-nuScenes.

| Method | Aux. Sup. | mIoU (%) ↑ | | | | IoU (%) ↑ | | | |
|---|---|---|---|---|---|---|---|---|---|
| | | 1s | 2s | 3s | Avg. | 1s | 2s | 3s | Avg. |
| OccWorld-S | None | 0.28 | 0.26 | 0.24 | 0.26 | 5.05 | 5.01 | 4.95 | 5.00 |
| OccWorld-T | Semantic LiDAR | 4.68 | 3.36 | 2.63 | 3.56 | 9.32 | 8.23 | 7.47 | 8.34 |
| OccWorld-D | 3D Occ | 11.55 | 8.10 | 6.22 | 8.62 | 18.90 | 16.26 | 14.43 | 16.53 |
| OccLLAMA-F | 3D Occ | 10.34 | 8.66 | 6.98 | 8.66 | **25.81** | **23.19** | **19.97** | **22.99** |
| PreWorld | 3D Occ | 11.69 | 8.72 | 6.77 | 9.06 | 23.01 | 20.79 | 18.84 | 20.88 |
| PreWorld+ | 2D Labels & 3D Occ | 12.27 | 9.24 | 7.15 | 9.55 | 23.62 | 21.62 | 19.63 | 21.62 |
| **STM4D (Ours)** | 2D Labels & 3D Occ | **12.48** | **9.41** | **7.60** | **9.83** | 23.46 | 21.04 | 19.51 | 21.33 |

resolution ($512 \times 512 \times 40$ grid). We evaluate the following four tasks: (1) Inflated GMO, (2) Inflated GMO with GSO, (3) Fine-grained GMO with GSO, and (4) Fine-grained GMO prediction. Performance is measured using the $IoU_c$, $IoU_f(2s)$, and $\tilde{IoU}_f$ metrics (Ma et al., 2024a).

## 4.2 Implementation and Training Details.

Each benchmark has distinct task definitions and evaluation metrics. We adapt our implementation accordingly to ensure fair comparisons with prior work. Multi-view 2D segmentation labels are auto-generated by Segment Anything (SAM) (Kirillov et al., 2023), avoiding manual annotation. See OccNerf (Zhang et al., 2023a) for details. All experiments were run on 8 NVIDIA RTX6000 Ada GPUs.

**Occ3D-nuScenes Benchmark:** Following Preworld (Li et al., 2025), we adopt BEV-Stereo's (Li et al., 2023b) architecture (2D backbone + lifting module) with $512 \times 1408$ input resolution. Training uses Adam optimizer (LR=$1 \times 10^{-4}$, batch size=8) for 18 epochs. The 2D branch employs cross-entropy loss (weight=1.0), while the 3D branch combines Focal loss, Lovász-Softmax loss, and scene-class affinity losses (each weight=1.0). For the 3D branch, we implement curriculum learning with dynamic frame prediction:

$$\text{future\_intervals}(e) = \begin{cases} \{0, 1\} & e \leq 4 \\ \{n \in \mathbb{N} \mid 0 \leq n \leq \min(\lfloor \frac{e-3}{2} \rfloor, 5)\} & \text{otherwise} \end{cases} \quad (10)$$

**Cam4DOcc Benchmark:** Building on Cam4DOcc (Ma et al., 2024a), we maintain their 2D backbone and lifting module but introduce our 2DST, 3DST, and STIM modules while removing the flow prediction branch. Input resolution is reduced to $448 \times 896$. Training configuration: 24 epochs, batch size 8, Adam ($1 \times 10^{-4}$ LR), with cross-entropy loss for both 2D segmentation and occupancy prediction (weights=1.0).

## 4.3 Main Results

**4D Occupancy Forecasting Results On Occ3D-nuScenes.** We compare 4D occupancy forecasting results on the Occ3D-nuScenes dataset of the proposed STM4D method and SOTA methods OccWorld (Zheng et al., 2025), OccLLaMA (Wei et al., 2024), PreWorld (Li et al., 2025), and PreWorld+ (the full method of Li et al. (2025)), in Table 1. STM4D outperforms all comparisons in terms of Avg. mIoU, indicating its strong power of spatio-temporal modeling for the 4D occupancy forecasting tasks. IoU is not a core metric, and further explanations will be provided in Appendix A.1. Notably, STM4D's advantage over the SOTA (PreWorld+) increases with time, gaining +0.45 mIoU at 3s versus +0.21 mIoU at 1s, underscoring its superior long-term forecasting capability. This stems from the powerful spatio-temporal modeling enabled by the synergistic interaction between our dedicated 2D and 3D modules.

**4D Occupancy Forecasting Results On Cam4DOcc.** We conduct comprehensive comparisons with OpenOccupancy-C (Wang et al., 2023b), SPC (Wei et al., 2023; Luo et al., 2023; Zhu et al., 2021), PowerBEV-3D (Li et al., 2023a), EfficientOCF (Xu et al.), OccProphet (Chen et al., 2025) and the OCFNet baseline (Ma et al., 2024a) on Cam4DOcc benchmark (Ma et al., 2024a). Experimental results demonstrate that our approach demonstrates consistent performance improvements for all

Table 2: Result on forecasting inflated GMO and fine-grained GSO on Cam4DOcc.

| Method | $IoU_c$ | | | $IoU_f$ (2 s) | | | $\tilde{IoU}_f$ |
|---|---|---|---|---|---|---|---|
| | GMO | GSO | mean | GMO | GSO | mean | GMO |
| OpenOccupancy-C | 13.53 | 16.86 | 15.20 | 12.67 | 17.09 | 14.88 | 12.97 |
| SPC | 1.27 | 3.29 | 2.28 | failed | 1.40 | – | failed |
| PowerBEV-3D | 23.08 | – | – | 21.25 | – | – | 21.86 |
| OCFnet | 29.84 | 17.72 | 23.78 | 25.53 | 17.81 | 21.67 | 26.53 |
| EfficientOCF | 32.12 | 22.97 | 27.55 | 26.16 | 23.23 | 24.70 | 28.53 |
| OccProphet | 33.61 | 24.18 | 28.89 | 26.45 | **24.19** | 25.32 | 28.74 |
| **STM4D (Ours)** | **33.92** | **24.75** | **29.34** | **26.83** | 24.15 | **25.49** | **29.34** |

Table 3: Comparison of different methods on forecasting inflated GMO on Cam4DOcc.

| Method | $IoU_c$ | $IoU_f$ (2 s) | $\tilde{IoU}_f$ |
|---|---|---|---|
| OpenOccupancy-C | 12.17 | 11.45 | 11.74 |
| SPC | 1.27 | failed | failed |
| PowerBEV-3D | 23.08 | 21.25 | 21.86 |
| OCFnet | 31.30 | 26.82 | 27.98 |
| EfficientOCF | 34.13 | 26.05 | 28.71 |
| OccProphet | 34.36 | 26.94 | 29.15 |
| **STM4D (Ours)** | **34.70** | **28.21** | **30.92** |

Table 4: Ablation study on the quantity of 3D supervision scenes (total of 700).

| Labels | | Avg. mIoU (%) ↑ | |
|---|---|---|---|
| 3D | 2D | PreWorld+ | STM4D |
| 150 | 700 | 25.02 | **25.71** |
| 450 | 700 | 33.37 | **34.31** |
| 700 | 700 | 34.69 | **35.18** |

tasks. For the inflated GMO and fine-grained GSO tasks (Table 2), we achieve a gain $0.45$ (29.34 vs 28.89) in $IoU_c$, a $0.17$ improvement (25.49 vs 25.32) in $IoU_f(2s)$, and a $0.60$ improvement (29.34 vs 28.74) in $\tilde{IoU}_f$. For inflated GMO prediction (Table 3), we observe a $0.34$ improvement (34.70 vs 34.36) in $IoU_c$, a $1.27$ increase (28.21 vs 26.94) in $IoU_f(2s)$, and a $1.77$ gain (30.92 vs 29.15) in $\tilde{IoU}_f$. Our method exhibits superior performance gains in future-frame over current-frame predictions, highlighting its strength in long-term forecasting. Additional experiments on fine-grained tasks are provided in Appendix A.2.

**Visualization.** Figure 4 compares qualitative results of our STM4D, PreWorld+ (Li et al., 2025), and OccWorld-D (Zheng et al., 2025), with each row showing predictions at 0.5s intervals over 3 seconds. Our method generates more accurate predictions: it correctly identifies "construction vehicle" (vs. "truck" misclassification by others) and consistently tracks the bottom "car" with precise motion trajectory, while others lose tracking or predict incorrect downward motion. These results demonstrate STM4D's superior temporal dynamic modeling in resolving motion ambiguities.

## 4.4 ABLATION STUDY

**Module Ablations.** We conduct systematic ablation studies to evaluate the contributions of key components in our STM4D framework, with results summarized in Table 5. Using PreWorld with 3D-only supervision as our baseline (Row 1, 8.16), we observe that incorporating the 2DST module improves performance to 8.45, demonstrating the importance of 2D spatio-temporal modeling. Removing the lstm module and retaining only the autoregressive lstm in 3DST achieved 8.51 mIoU, while using the full 3DST module (autoregressive lstm + lstm module) alone achieves 8.89, confirming the effectiveness of our 3D temporal modeling design. The combination of both 2DST and 3DST reaches 9.64, indicating their complementary benefits in multi-view temporal reasoning. Our full model, incorporating all three modules, achieves the best performance of 9.83, demonstrating that the STIM-mediated 2D-3D interactions further refine temporal feature representations and contribute to the state-of-the-art performance.

**Semi-3D Supervision Ablations.** We follow PreWorld+'s semi-supervised setup and reduce 3D supervision from 700 to 450 and 150 scenes to evaluate performance under limited 3D labels. As Table 4 shows, both methods decline with fewer 3D labels, yet ours consistently outperforms

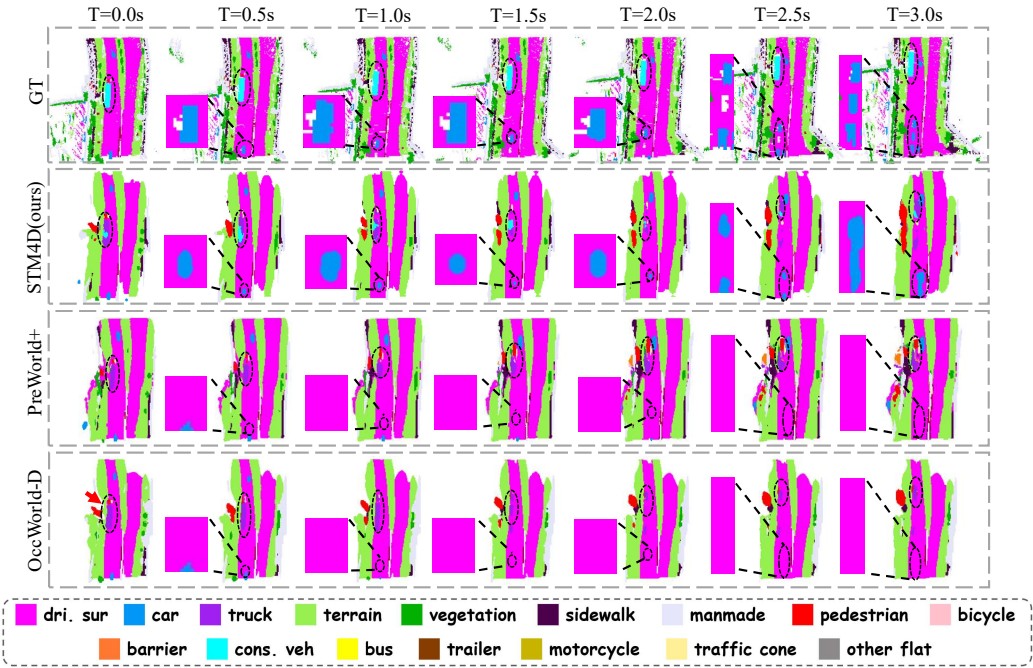

Figure 4: The visualization of the 4D occupancy forecasting results. The black circles indicate that our method can achieve more accurate results than SOTA PreWorld+ (full method of Li et al. (2025)) and OccWorld-D (Zheng et al., 2025).

Table 5: Ablation study on method modules.

| 3DST(full) | 3DST(wt.lstm) | 2DST | STIM | Avg. mIoU (%) ↑ |
|------------|---------------|------|------|-----------------|
| ✗ | ✗ | ✗ | ✗ | 8.16 |
| ✗ | ✗ | ✓ | ✗ | 8.45 |
| ✗ | ✓ | ✗ | ✗ | 8.51 |
| ✓ | ✗ | ✗ | ✗ | 8.89 |
| ✓ | ✗ | ✓ | ✗ | 9.64 |
| ✓ | ✗ | ✓ | ✓ | **9.83** |

Table 6: Ablation study on frame count for 2D prediction.

| Frame Count | Avg. mIoU (%) ↑ |
|-------------|-----------------|
| 3 | 9.45 |
| 5 | 9.61 |
| 7 | **9.83** |

PreWorld+, demonstrating the superiority of our spatio-temporal modeling strategy. Notably, our 2D spatio-temporal branch acts as effective regularization that compensates for reduced 3D supervision.

**Ablation Study on Frame Count for 2D Prediction.** We ablate the number of multi-view segmentation maps predicted by the 2DST module (0.5s inter-frame interval). As shown in Table 6, performance improves from 9.45 mIoU (3 frames) to 9.61 mIoU (5 frames). Best results (9.83 mIoU at 0–3s) occur when the 2D prediction length matches the 3DST output (7 frames), confirming that longer horizons in the 2D branch enhance temporal prior aggregation and boost performance.

## 5 CONCLUSION

We focus on a novel perspective that integrates simultaneous 2D and 3D spatio-temporal modeling and leverages temporal cues from 2D multi-view image sequences to improve 4D occupancy prediction, proposing the STM4D model as our solution. STM4D consists of the 3D spatio-temporal modeling module 3DST as well as the 2D spatio-temporal modeling module 2DST. Besides, we also propose an extra 2D and 3D spatio-temporal interaction module STIM for further enhancing the spatio-temporal fusion of the 2D and 3D features. Experimental results demonstrate the effectiveness of our method in 4D occupancy forecasting, particularly in long-term prediction scenarios. The proposed spatio-temporal modeling framework provides a new perspective for further enhancing the performance of autonomous driving systems.

ETHICS STATEMENT

This work presents a novel framework for 4D occupancy forecasting, a critical task for enhancing the safety and situational awareness of autonomous driving systems. Our research utilizes publicly available benchmarks, Occ3D-nuScenes (Tian et al., 2023) and Cam4DOcc (Ma et al., 2024a), which consist of data collected in accordance with their respective licenses and ethical guidelines. These datasets contain no personally identifiable information. We foresee the primary societal benefit of this work to be the advancement of reliable perception systems for autonomous vehicles, potentially leading to reduced traffic accidents. We are not aware of any direct negative societal impacts of our method itself, but we acknowledge the broader ethical considerations common to all autonomous driving technologies, such as decision-making in edge cases and the potential for job displacement. We encourage the responsible development and deployment of such technologies, underscored by rigorous testing and transparent regulatory frameworks.

REPRODUCIBILITY STATEMENT

To ensure reproducibility, we detail our methodology and experiments. The STM4D architecture is described in Section 3, including the 3DST (Sec. 3.2), 2DST (Sec. 3.3), and STIM (Sec. 3.4) modules. Implementation details (e.g., network architectures, hyperparameters, training configurations) for the Occ3D-nuScenes and Cam4DOcc benchmarks are provided in Section 4.2, along with the process for generating 2D segmentation labels using SAM (Kirillov et al., 2023). The datasets are public. Code, models, and training logs will be released upon acceptance. Experiments used 8 NVIDIA RTX 6000 Ada GPUs; scripts will be provided to facilitate replication.

LLM USAGE STATEMENT

Large Language Models (LLMs), such as ChatGPT, were used to assist with language polishing, grammar correction, and improving the clarity of the manuscript. All technical ideas, model designs, experiments, and analyses were conceived and executed by the authors. The LLM did not generate novel research content or influence the reported scientific results.

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

# A  APPENDIX

## A.1  DETAILED EXPLANATION

**About the mIoU/IoU Performance Discrepancy.** It is noteworthy that in Table 1, our STM4D method achieves comprehensive superiority in the mIoU metric, yet it does not attain comparable performance in terms of IoU. A similar trend is observed for PreWorld (Li et al., 2025) (second to last row).

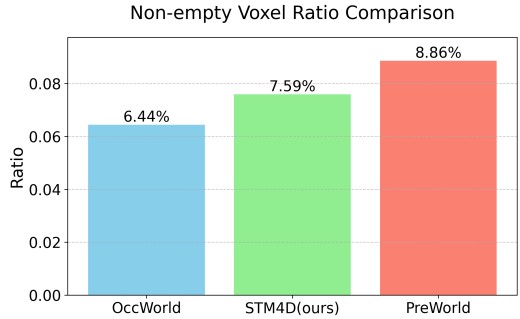

We emphasize that mIoU is widely recognized as the primary evaluation metric for this task, whereas IoU is not considered a core criterion. By jointly modeling 2D and 3D spatio-temporal information, STM4D prioritizes trajectory and semantic consistency of key dynamic objects (e.g., vehicles and pedestrians), which may come at the cost of some scene coverage. As shown in Figure 5, our method achieves a lower proportion of non-empty voxels than PreWorld (Li et al., 2025) but higher than OccWorld (Zheng et al., 2025), which results in a lower IoU than PreWorld but higher than OccWorld.

Figure 5: The non-empty voxels voxel coverage predicted by our STM4D is lower than that of PreWorld but higher than that of OccWorld, resulting in a correspondingly lower IoU than PreWorld but a higher one than OccWorld.

As illustrated in Figure 6, although STM4D produces scenes with slightly lower completeness compared to PreWorld, it yields more accurate predictions for moving objects and road structures. Therefore, the observed relative performance in IoU should not be construed as a model limitation. Instead, the superior mIoU results reflect a deliberate and meaningful design trade-off, one that better aligns with the practical needs of autonomous driving systems.

**The setting of the LSTM implementation in the 3DST module.** In the 3D Spatio-Temporal (3DST) module, our LSTM architecture fundamentally extends the ConvLSTM (Shi et al., 2015) framework by replacing all 2D convolutional operations with 3D convolutions to natively process 3D voxel data while preserving temporal modeling capabilities. Specifically, the autoregressive LSTM employs a sliding window mechanism with a fixed size of 2, utilizing two consecutive historical time steps to predict the subsequent time step, effectively balancing computational efficiency and temporal context utilization. The detailed configuration adopts a single hidden layer. The hidden state dimension of the autoregressive LSTM is set to 16, while the others are set to 32, with $3 \times 3 \times 3$ kernel sizes used for all convolutional operations.

**The details of 2DST module.** For the 2D CNNs in 2DST (Figure 2), the input to the module consists of multi-view 2D image features with dimensions $(N, T, C, h, w)$. These features are then concatenated along the channel dimension to form image features of shape $(N, T \times C, h, w)$. Subsequently, we employ DenseNet (Huang et al., 2017) to extract multi-scale features. Specifically, through this operation, the original image features are downsampled to $1/4$, $1/8$, and $1/16$ of the original feature resolution, where the $1/16$ resolution is $(8, 22)$ with 256 feature channels. These features are then aligned to the $1/16$ resolution and concatenated together. In this process, the concatenated features serve as a bridge linking past frames and historical frames, containing rich temporal information from the 2D sequence. The concatenated features are flattened and used as input to STIM, forming the fused features. Simultaneously, the concatenated features also undergo channel expansion and are reshaped to align with future frames. They are then processed by 3D transposed convolutions to progressively upsample to the target segmentation size while refining temporal relationships. Finally, a Conv3D layer further refines the output to produce future image features.

**The details of STIM module.** Since the fused features from 2DST, which contain rich 2D temporal information with shape $(B, L, C)$, are obtained by flattening multi-view feature maps with sufficiently small resolution $(8 \times 22)$, directly applying global cross-attention to the fused features does not incur significant computational overhead. The specific procedure of our STIM (Figure 2) is as follows:

First, we initialize a query tensor of size $(200, 200, 256)$. Using the fused features as keys and values, we apply cross-attention to obtain a BEV feature of size $(200, 200, 256)$. Next, an MLP is used to predict the height feature distribution, which is then reshaped to yield 3D features $(C, 200, 200, 16)$. Finally, these 3D features are aligned to future frames through replication, then concatenated with the refined spatio-temporal features and fed into an LSTM. The output of the LSTM is then combined with the refined spatio-temporal features to produce the enhanced features, which serve as the final output of the STIM module.

**The reason for using different methods for comparison for two datasets.** The two datasets employ fundamentally different task formulations and evaluation metrics: Occ3D-nuScenes evaluates 4D semantic occupancy prediction across 18 classes using mIoU/IoU metrics, while Cam4DOcc progressively assesses performance through four difficulty tiers using GSO/GMO IoU measurements. Due to these inherent methodological differences, most works (e.g., OccWorld, OccLLAMA, and PreWorld on Occ3D-nuScenes; OCFnet, PowerBEV-3D, and SPC on Cam4DOcc) report results on only one dataset, making direct cross-dataset comparisons methodologically unsound.

## A.2 MORE EXPERIMENTS

**3D Occupancy Prediction Results On Occ3D-nuScenes.** We conduct a comprehensive evaluation of 3D occupancy prediction performance on the Occ3D-nuScenes benchmark, comparing our proposed STM4D framework against current state-of-the-art methods in Table 7. To adapt our spatio-temporal forecasting architecture for the static occupancy task, we configure both the 3DST and 2DST modules with a prediction length of 1, focusing exclusively on reconstructing the current scene structure. The results demonstrate the compelling effectiveness of our approach. STM4D achieves state-of-the-art performance with an overall mIoU of 35.18, surpassing all competing methods by a significant margin. Notably, our method outperforms strong baselines such as PreWorld+ (34.69 mIoU) and OccFlowNet (33.86 mIoU). A fine-grained analysis reveals that STM4D achieves top performance in 9 out of 17 semantic categories, exhibiting particular strength in challenging classes including *barrier* (45.73), *car* (50.07), and *drivable surface* (67.55). The comprehensive performance gains, evident in both overall and category-specific metrics, underscore STM4D's exceptional capability in geometric reasoning and structural understanding. These results validate that our spatio-temporal modeling paradigm, even when specialized for static scene comprehension, effectively captures complex 3D structures and semantic relationships, yielding more accurate occupancy predictions than methods specifically designed for this task.

Table 7: **3D occupancy prediction performance on the Occ3D-nuScenes dataset.** GT represents the type of labels used during training. The best results are represented by **bold**.

| Method | GT | others | barrier | bicycle | bus | car | cons. veh | motorcycle | pedestrian | traffic cone | trailer | truck | dri. sur | other flat | sidewalk | terrain | manmade | vegetation | mIoU (%) |
|---|---|---|---|---|---|---|---|---|---|---|---|---|---|---|---|---|---|---|---|
| SelfOcc (Huang et al., 2024) | 2D | 0.00 | 0.15 | 0.66 | 5.46 | 12.54 | 0.00 | 0.80 | 2.10 | 0.00 | 0.00 | 8.25 | 55.49 | 0.00 | 26.30 | 26.54 | 14.22 | 5.60 | 9.30 |
| OccNeRF (Zhang et al., 2023a) | 2D | 0.00 | 0.83 | 0.82 | 5.13 | 12.49 | 3.50 | 0.23 | 3.10 | 1.84 | 0.52 | 3.90 | 52.62 | 0.00 | 20.81 | 24.75 | 18.45 | 13.19 | 9.53 |
| RenderOcc (Pan et al., 2024b) | 2D | 5.69 | 27.56 | 14.36 | 19.91 | 20.56 | 11.96 | 12.42 | 12.14 | 14.34 | 20.81 | 18.94 | 68.85 | 33.35 | 42.01 | 43.94 | 17.36 | 22.61 | 23.93 |
| OccFlowNet (Boeder & Risse, 2025) | 2D | 1.60 | 27.50 | 26.00 | 34.00 | 32.00 | 20.40 | 25.90 | 18.60 | 20.20 | 26.00 | 28.70 | 62.00 | 27.20 | 37.80 | 39.50 | 29.00 | 26.80 | 28.42 |
| MonoScene (Cao & De Charette, 2022) | 3D | 1.75 | 7.23 | 4.26 | 4.93 | 9.38 | 5.67 | 3.98 | 3.01 | 5.90 | 4.45 | 7.17 | 14.91 | 6.32 | 7.92 | 7.43 | 1.01 | 7.65 | 6.06 |
| TPVFormer (Huang et al., 2023) | 3D | 7.22 | 38.90 | 13.67 | 40.78 | 45.90 | 17.23 | 19.99 | 18.85 | 14.30 | 26.69 | 34.17 | 55.65 | 35.47 | 37.55 | 30.70 | 19.40 | 16.78 | 27.83 |
| BEVDet (Huang et al., 2021) | 3D | 4.39 | 30.31 | 0.23 | 32.26 | 34.47 | 12.97 | 10.34 | 10.36 | 6.26 | 8.93 | 23.65 | 52.27 | 24.61 | 26.06 | 22.31 | 15.04 | 15.10 | 19.38 |
| OccFormer (Zhang et al., 2023b) | 3D | 5.94 | 30.29 | 12.32 | 34.40 | 39.17 | 14.44 | 16.45 | 17.22 | 9.27 | 13.90 | 26.36 | 50.99 | 30.96 | 34.66 | 22.73 | 6.76 | 6.97 | 21.93 |
| BEVFormer (Li et al., 2024) | 3D | 5.85 | 37.83 | 17.87 | 40.44 | 42.43 | 7.36 | 23.88 | 21.81 | 20.98 | 22.38 | 30.70 | 55.35 | 28.36 | 36.00 | 28.06 | 20.04 | 17.69 | 26.88 |
| RenderOcc (Pan et al., 2024b) | 2D+3D | 4.84 | 31.72 | 10.72 | 27.67 | 26.45 | 13.87 | 18.20 | 17.67 | 17.84 | 21.19 | 23.25 | 63.20 | 36.42 | 46.21 | 44.26 | 19.58 | 20.72 | 26.11 |
| CTF-Occ (Tian et al., 2023) | 3D | 8.09 | 39.33 | 20.56 | 38.29 | 42.24 | 16.93 | 24.52 | 22.72 | 21.05 | 22.98 | 31.11 | 53.33 | 33.84 | 37.98 | 33.23 | 20.79 | 18.00 | 28.53 |
| SparseOcc (Tang et al., 2024) | 3D | - | - | - | - | - | - | - | - | - | - | - | - | - | - | - | - | - | 30.90 |
| OccFlowNet (Boeder & Risse, 2025) | 2D+3D | 8.00 | 37.60 | 26.00 | 42.10 | 42.50 | 21.60 | 29.20 | 22.30 | 25.70 | 29.70 | 34.40 | 64.90 | 37.20 | 44.30 | 43.20 | **34.30** | **32.50** | 33.86 |
| PreWorld (Li et al., 2025) | 3D | 10.83 | 44.13 | 26.35 | 42.16 | 46.15 | 22.92 | 28.86 | 26.89 | 26.44 | 28.29 | 34.43 | 65.67 | 35.91 | 41.09 | 37.41 | 30.16 | 29.54 | 33.95 |
| PreWorld+ (Li et al., 2025) | 2D+3D | 11.81 | 45.01 | 26.29 | **43.32** | 47.71 | 24.23 | **31.29** | 27.41 | 27.68 | 30.62 | **35.64** | 63.71 | **37.27** | 41.20 | 37.54 | 29.36 | 29.70 | 34.69 |
| **STM4D (Ours)** | 2D+3D | **12.63** | **45.73** | **27.73** | 33.92 | **50.07** | **26.98** | 27.96 | **28.69** | **27.88** | **30.74** | 35.18 | **67.55** | 35.74 | 39.52 | 42.82 | 32.76 | 32.13 | **35.18** |

**Fine-grained GMO prediction task on Cam4DOcc.** We conduct comprehensive comparisons with OpenOccupancy-C (Wang et al., 2023b), SPC (Wei et al., 2023; Luo et al., 2023; Zhu et al., 2021), PowerBEV-3D (Li et al., 2023a), OccProphet (Chen et al., 2025) and the OCFNet baseline (Ma et al., 2024a) on Cam4DOcc benchmark (Ma et al., 2024a). On the fine-grained GMO task (Table 8), our proposed STM4D also establishes a new state-of-the-art performance, achieving the highest scores across all evaluation metrics. As shown in Table 8, STM4D attains an $IoU_c$ of 11.98,

Table 8: Comparison of different methods on forecasting fine-grained GMO on Cam4DOcc.

| Method | $\text{IoU}_c$ | $\text{IoU}_f$ (2 s) | $\tilde{\text{IoU}}_f$ |
|---|---|---|---|
| OpenOccupancy-C | 10.82 | 8.02 | 8.53 |
| SPC | 5.85 | 1.08 | 1.12 |
| PowerBEV-3D | 5.91 | 5.25 | 5.49 |
| OCFnet | 11.45 | 9.68 | 10.10 |
| OccProphet | 15.38 | 10.69 | 11.98 |
| **STM4D (Ours)** | **11.98** | **11.03** | **12.54** |

Table 9: LSTM replacement comparison.

| **Method** | **mIoU (%) ↑** | | | |
|---|---|---|---|---|
| | 1s | 2s | 3s | Avg. |
| Transformer | 12.16 | 9.08 | 7.20 | 9.48 |
| GRU | 12.21 | 9.17 | 7.16 | 9.51 |
| LSTM | **12.48** | **9.41** | **7.60** | **9.83** |

Table 10: Comparison of different methods on forecasting fine-grained GMO and fine-grained GSO simultaneously on Cam4DOcc.

| Method | $\text{IoU}_c$ | | | $\text{IoU}_f$ (2 s) | | | $\tilde{\text{IoU}}_f$ |
|---|---|---|---|---|---|---|---|
| | GMO | GSO | mean | GMO | GSO | mean | GMO |
| OpenOccupancy-C | 9.62 | 17.21 | 13.42 | 7.41 | 17.30 | 12.36 | 7.86 |
| SPC | 5.85 | 3.29 | 4.57 | 1.08 | 1.40 | 1.24 | 1.12 |
| PowerBEV-3D | 5.91 | – | – | 5.25 | – | – | 5.49 |
| OCFnet | 11.02 | 17.79 | 14.41 | 9.20 | 17.83 | 13.52 | 9.66 |
| OccProphet | 13.71 | 24.42 | 19.06 | 9.34 | 24.56 | 16.95 | 10.33 |
| **STM4D (Ours)** | **14.03** | **24.81** | **19.42** | **9.53** | **25.06** | **17.30** | **10.89** |

$\text{IoU}_f(2s)$ of 11.03, and $\tilde{\text{IoU}}_f$ of 12.54, outperforming all competing methods. Specifically, STM4D shows significant improvements over previous state-of-the-art approaches: a 22.3% increase in $\tilde{\text{IoU}}_f$ compared to OccProphet (Chen et al., 2025) (11.98 vs. 10.10), a 24.2% improvement in $\text{IoU}_f(2s)$ over OCFnet (Ma et al., 2024a) (11.03 vs. 9.68), and substantial gains over OpenOccupancy-C (Wang et al., 2023b) across all metrics. These consistent improvements highlight STM4D's superior capability in accurately capturing and predicting complex dynamic object trajectories, particularly in challenging fine-grained scenarios where precise motion forecasting is essential.

**Fine-grained GMO and fine-grained GSO prediction task on Cam4DOcc.** Our proposed framework, STM4D, establishes new state-of-the-art performance on the challenging task of fine-grained future scene forecasting, significantly outperforming existing methods in predicting both General Moving Objects (GMO) and General Static Occupancy (GSO). The quantitative results, summarized in Table 10, demonstrate our method's comprehensive superiority. STM4D achieves the highest scores across all critical evaluation metrics: it attains a mean $\text{IoU}_c$ of 19.42 (with a decomposition of 14.03 for GMO and 24.81 for GSO), an $\text{IoU}_f(2s)$ of 17.30 (9.53 for GMO and 25.06 for GSO), and a $\tilde{\text{IoU}}_f$ of 10.89 specifically for GMO forecasting.Notably, the performance gain in $\tilde{\text{IoU}}_f$ for GMO—a key metric for evaluating the forecasting of dynamic agents—represents a substantial improvement of 5.4% over the previous leading method, OccProphet (Chen et al., 2025), and a marked 12.7% increase over OCFnet (Ma et al., 2024a). These results robustly demonstrate STM4D's superior capability in capturing and modeling the complex spatiotemporal dependencies inherent in dynamic autonomous driving scenarios, ultimately leading to more accurate and reliable long-horizon predictions.

**Superiority of LSTM in Spatiotemporal Forecasting.** To select the most suitable sequence modeling architecture for our task, we conduct comprehensive experiments comparing several classical and well-established sequence modeling architectures, including ConvLSTM (Shi et al., 2015) (here uniformly referred to as LSTM), Transformer (Vaswani et al., 2017), and GRU (Chung et al., 2014) variants. Based on the comparative results in Table 9, LSTM achieves the highest mIoU across all evaluated time horizons, scoring 12.48, 9.41, and 7.60 at 1s, 2s, and 3s, respectively, with an average of 9.83. While GRU slightly exceeds Transformer on average performance (9.51 vs. 9.48), both exhibit a noticeable decline in accuracy over time. These results demonstrate LSTM's superior capability in modeling long-term temporal dependencies for the presented task, providing strong empirical evidence that its gated memory mechanism and gradient preservation properties offer

Table 11: Comparison of model complexity: number of parameters $N_p$, GPU memory, and inference FPS.

| Dataset | Method | $N_p$ (Million) | Memory (G) | FPS |
|---|---|---|---|---|
| Occ3D-nuScenes | OccWorld-D (Zheng et al., 2025) | - | 23 | 2.3 |
| | PreWorld (Li et al., 2025) | 121 | 29 | 4.1 |
| | Ours | 155 | 33 | 4.2 |
| Cam4DOcc | OCFnet (Ma et al., 2024a) | 370 | 57 | 1.7 |
| | OccProphet (Chen et al., 2025) | 82 | 24 | 4.5 |
| | Ours | 81 | 22 | 4.5 |

distinct advantages for capturing complex spatiotemporal relationships—particularly in challenging forecasting scenarios that demand sustained temporal coherence.

### A.3 METHOD COMPUTATION COMPARISON

To ensure fair comparisons across different dataset domains where methods employ varying 2D image backbones and 2D-to-3D lifting modules—and not all approaches report results on both benchmarks—we conduct separate evaluations per dataset. We analyze computational efficiency in Table 11, comparing model parameters, GPU memory usage, and inference speed measured on an NVIDIA RTX 6000 Ada GPU. Our implementation follows established configurations: on Occ3D-nuScenes (Tian et al., 2023), we adopt PreWorld's (Li et al., 2025) backbone and lifting module design, while on Cam4DOcc (Ma et al., 2024a), we align with OCFnet's architecture.

Our model achieves superior efficiency with 81M parameters on Cam4DOcc (compared to OCFnet's 370M and OccProphet's 82M) and 155M parameters on Occ3D-nuScenes (compared to PreWorld's 121M), demonstrating significant parameter efficiency. Training memory consumption remains moderate at 22GB on Cam4DOcc and 33GB on Occ3D-nuScenes, while maintaining competitive inference speeds of 4.5 FPS on both datasets. This optimized performance-to-computation ratio—achieving the highest FPS while maintaining parameter efficiency—demonstrates our method's practical advantages for real-world deployment scenarios.

### A.4 MORE VISUALIZATION

We also visualize qualitative results from our STM4D, PreWorld+ (Li et al., 2025), and OccWorld-D (Zheng et al., 2025), with each row displaying occupancy predictions at 0.5-second intervals over a 3-second horizon. The black circles highlight the regions where our STM4D demonstrates superior predictive performance compared to other methods.

As shown in Figure 6, our STM4D method more accurately predicts the U-shaped structure of the drivable area while maintaining its structural consistency over the 3-second horizon. Additionally, our approach effectively detects parked vehicles on the roadside and retains them consistently throughout the prediction duration. In contrast, while other methods achieve satisfactory performance on initial frames, they exhibit significant degradation in subsequent predictions, often failing to maintain road topology and completely losing track of vehicle instances. These results demonstrate the model's strong capability in capturing complex spatial structures and maintaining robust temporal consistency for both static and dynamic objects in long-horizon occupancy prediction tasks.

As illustrated in Figure 7, our STM4D method accurately predicts both the road structure and vehicles on the roadside while consistently maintaining their state over time. In contrast, although other methods achieve reasonable accuracy in the initial frames, the road structure becomes severely corrupted in later frames, and roadside vehicles are completely lost. These observations further demonstrate the superior temporal consistency and robustness of our approach in long-term occupancy forecasting.

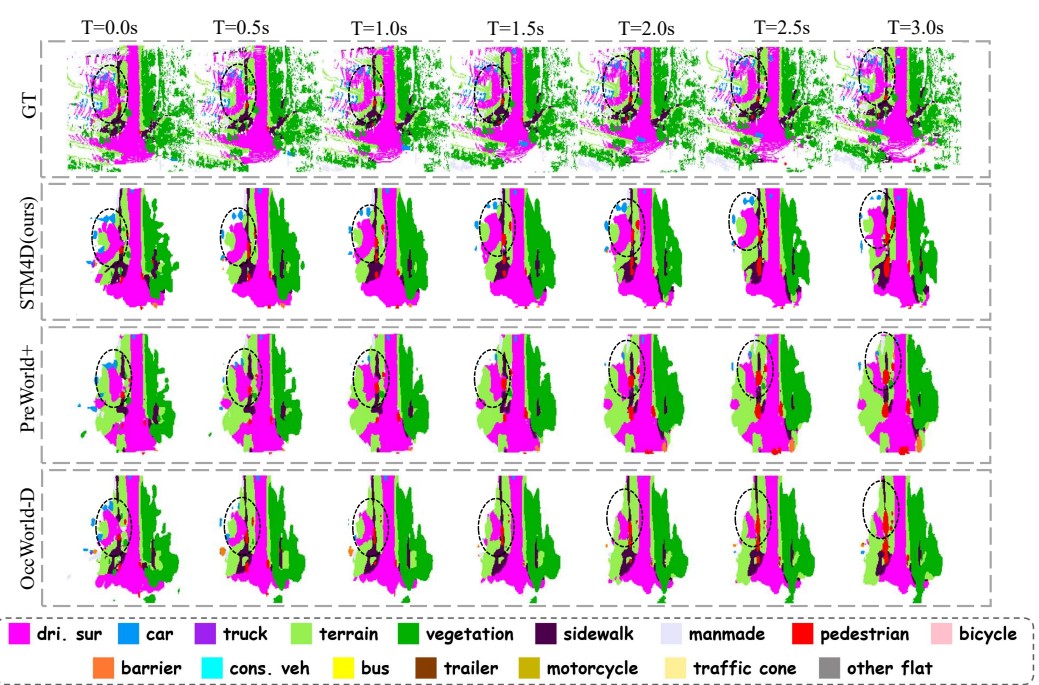

Figure 6: The visualization of the 4D occupancy forecasting results. The black circles indicate that our method can achieve more accurate results than SOTA PreWorld+ (full method of Li et al. (2025)) and OccWorld-D (Zheng et al., 2025).

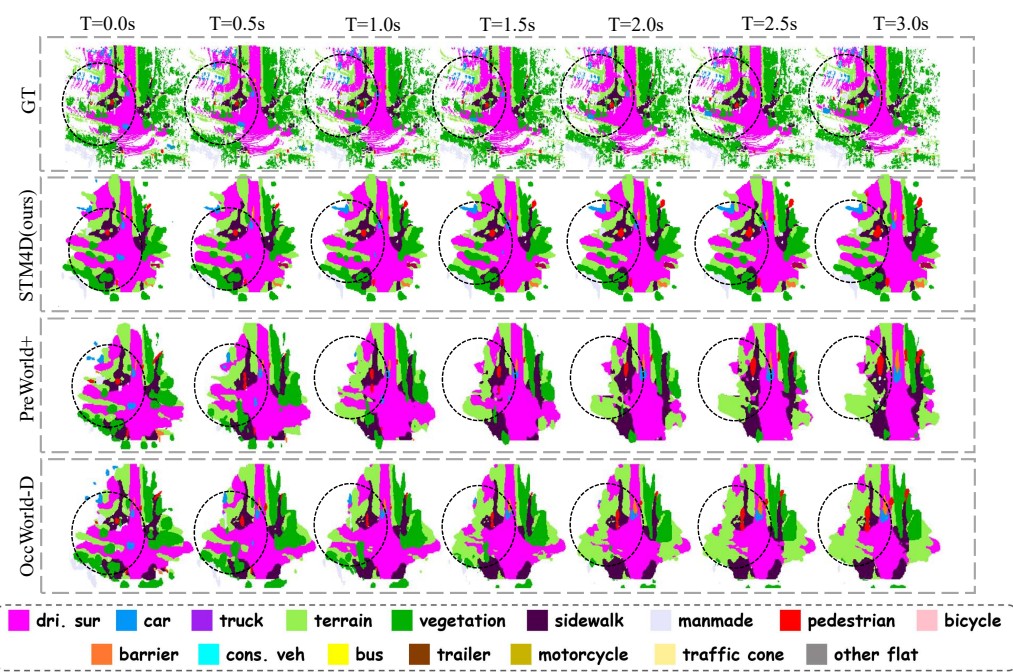

Figure 7: The visualization of the 4D occupancy forecasting results. The black circles indicate that our method can achieve more accurate results than SOTA PreWorld+ (full method of Li et al. (2025)) and OccWorld-D (Zheng et al., 2025).

