# OpenReview forum: "STM4D: 4D Occupancy Forecasting with 2D and 3D Spatio-Temporal Modeling"
_ICLR.cc/2026/Conference — ICLR 2026 Conference Withdrawn Submission_

### Official Review · Reviewer_KpCi · 2025-10-30

**Soundness:** 2
**Presentation:** 3
**Contribution:** 2
**Rating:** 4
**Confidence:** 4

**Summary:**

STM4D is a framework for 4D occupancy forecasting that predicts future 3D semantic scenes from videos by jointly modeling spatiotemporal dynamics in both 2D image space and 3D voxel space through a Spatio-Temporal Interaction Module (STIM). The STIM integrates 2D feature representations with refined 3D features, enabling end-to-end training. Experiments conducted on the Occ3D-nuScenes and Cam4DOcc benchmarks demonstrate competitive, and often state-of-the-art, performance.

**Strengths:**

This paper addresses an important problem—vision-only 4D occupancy forecasting. This topic holds significant practical relevance for autonomous driving. The paper is clearly written, and the diagrams are intuitive and easy to understand.

**Weaknesses:**

The experimental results are somewhat limited, as they are restricted to the Occ3D-nuScenes and Cam4DOcc datasets. Additional evaluations on more datasets (e.g., Lyft Level-5) will strengthen the validation of the proposed approach. While the joint 2D–3D design is reasonable, similar insights have already been explored and verified in prior occupancy-related literature. Compared with Cam4DOcc, this work represents an incremental improvement rather than a fundamentally novel formulation.

**Questions:**

1. Please add experiments and comparisons on the Lyft Level-5 to assess the performance.
(Lyft Level-5 Perception Dataset, 2019)
2. Please compare STM4D against EfficientOCF (CVPR 2025).
3. EfficientOCF explores spatiotemporal decoupling; what are the principal conceptual and architectural differences, and under what conditions would each be preferable?
4. The authors concatenate historical image features along the channel dimension. How do you mitigate spatial misalignment across views and time (e.g., due to ego-motion)? Is there any warping or learned alignment?
5. Since STIM relies on the quality of the generated BEV features, how do you ensure the geometric correctness of those features?
6. In Table 1, why does STM4D (supervised by 2D Labels & 3D Occ) underperform OccLLAMA-F (supervised only by 3D Occ) on Avg. IoU?
7. How does STM4D remain robust when objects become occluded over time?
8.Since 2D labels are auto-generated by SAM, how do you map SAM masks/classes to the occupancy taxonomy, and how do you reduce semantic drift between 2D labels and 3D occupancy classes?

---

> ### Author Response · Authors · 2025-11-29
> **Response to Reviewer KpCi**
>
> # Rebuttal
>
> ## Weaknesses
>
> **A1:** Our method, for the first time compared to previous approaches, introduces a multi-view 4D image segmentation task to enhance the performance of 4D occupancy. Unlike methods that rely on other 2D sequences, multi-view image segmentation labels are easier to acquire (e.g., via SAM) compared to other 2D annotations such as BEV segmentation. Moreover, the interaction between our STIM module and the 3D branch helps reduce the dependency on 3D annotations.
>
> ## Questions
>
>
> **A1:** We have supplemented the inflated GMO prediction results on the Lyft Level-5 dataset and conducted comparisons using the evaluation metrics from Cam4DOcc. As shown in the table below, our method continues to achieve leading performance.
>
> **Table:** Comparison of different methods on forecasting inflated GMO on Lyft Level-5.
>
> | Method | IoU$_c$ | IoU$_f$ (0.8s) | $\tilde{\text{IoU}}_f$ |
> |--------|---------|----------------|------------------------|
> | OpenOccupancy-C | 14.01 | 13.53 | 13.71 |
> | SPC | 1.42 | failed | failed |
> | PowerBEV-3D | 26.19 | 24.47 | 25.06 |
> | OCFnet | 36.41 | 33.56 | 34.60 |
> | EfficientOCF | 41.24 | 35.07 | 39.28 |
> | OccProphet | 43.38 | 37.92 | 40.11 |
> | **STM4D (Ours)** | **44.25** | **38.87** | **41.05** |
>
>
> **A2:** In our experiments, we maintained the same resolution for EfficientOCF as our method (448×800) and conducted comparisons using the evaluation metrics proposed in Cam4DOcc. Our results demonstrate that STM4D continues to achieve superior performance on the Cam4DOcc dataset for both the inflated GMO and fine-grained GSO combined task and the standalone inflated GMO task. We have revised Tables 2 and 3 in the manuscript to provide more intuitive comparisons.
>
>
> **A3:** EfficientOCF utilizes task-level spatiotemporal decoupling, where spatial decoupling combines BEV occupancy with height values to prevent 3D voxel redundancy, and temporal decoupling employs instance-aware associations to achieve improved 3D occupancy flow results. In contrast, our method performs implicit spatiotemporal decoupling at the feature level, without relying on additional annotations such as flow, BEV occupancy, or height values, demonstrating greater generality and broader applicability.
>
> Unlike methods that rely on other 2D sequences (e.g., BEV segmentation), multi-view image segmentation labels are easier to acquire (e.g., via SAM) compared to other 2D annotations such as BEV segmentation. Moreover, the interaction between our STIM module and the 3D branch helps reduce the dependency on 3D annotations. EfficientOCF's training process is computationally expensive, requiring substantial GPU memory, and relies heavily on 3D annotations. In contrast, our approach significantly reduces this dependency on 3D supervision.
>
>
>
> **A4:** Our approach addresses the practical constraint of unavailable future camera parameters at test time by forgoing explicit geometric operations like cross-view alignment or temporal warping. We propose a purely learning-based alternative that compresses multi-view features into a compact token space, where a unified representation is learned directly from data.
>
>
>
> **A5:** It is worth noting that our goal is to integrate the temporal and semantic information of the entire 2D sequence—from past to future—with the 3D branch. However, since future camera parameters are unavailable, we cannot ensure geometric consistency through projection. Instead, our parameter-free interaction enables the temporal and semantic information from the 2D branch to flow into the 3D branch in a purely learning-based manner.
>
>
> **A6:** A detailed explanation regarding this point has already been provided in Section A.1 of the supplementary material.
>
>
>
> **A7:** Our lifting module, adopted from BEVDet4D, incorporates features from adjacent frames when constructing the voxel representation for a given timestamp. This design ensures that an object can be detected as long as it is not occluded in every single frame.Regarding the mapping from 2D to 3D labels, we follow the methodology outlined in OccNeRF [1].
>
> [1] OccNeRF: Advancing 3D Occupancy Prediction in LiDAR-Free Environments

---

### Official Review · Reviewer_Y9Bu · 2025-10-31

**Soundness:** 3
**Presentation:** 2
**Contribution:** 2
**Rating:** 4
**Confidence:** 3

**Summary:**

The paper presents STM4D, a unified deep learning framework for 4D occupancy forecasting in autonomous driving scenarios. STM4D integrates two complementary spatio-temporal modeling modules: a 3D Spatio-Temporal (3DST) module for capturing volumetric dynamics; and a 2D Spatio-Temporal (2DST) module employing an auxiliary segmentation forecasting task to enforce temporal semantic consistency in the image space. The approach also introduces a Spatio-Temporal Interaction Module (STIM) to facilitate cross-modal interactions between the 2D and 3D representations without explicit reliance on camera parameters. Experiments on the Occ3D-nuScenes and Cam4DOcc benchmarks demonstrate state-of-the-art results, with ablations validating each component.

**Strengths:**

1. The paper conducts extensive evaluations on two leading benchmarks and sets the new SOTA.
2. The qualitative comparisons offer clear visual evidence that STM4D produces more temporally consistent and geometrically plausible predictions over multi-second horizons.

**Weaknesses:**

1. The framework design lacks detailed explanations. For instance, in the section on the 3DST module: What are the shapes of the intermediate features? Along which dimension are the 3D features of past and current frames concatenated? Why does the spatial encoder take the concatenated 3D features as a whole as input, yet produce spatial features for multiple future frames? Are these features consistent across frames, or is there some kind of autoregressive module involved? Similar questions also arise regarding the 2DST module, and we hope further clarifications can be provided in the subsequent rebuttal.
2. The ablation analysis could go deeper. For instance, the 3DST module employs two LSTM modules for temporal forecasting and feature refinement, based on the argument that this "enhances their temporal consistency." However, this design choice is insufficiently validated by the experimental results.
3. Despite the abundant set of modular designs incorporated into the model, the performance improvement over previous SOTA methods appears somewhat limited. For example, the average mIoU improvment on Occ3D-nuScenes is only about 3% compared to PreWorld+.

**Questions:**

1. Given the number of components in STM4D, it is notable that its FPS remains comparable to PreWorld. To better understand this efficiency, could the authors provide a detailed breakdown of the latency introduced by each module?
2. We note from Table 7 that STM4D does not achieve SOTA prediction performance across all semantic categories. Could the authors provide some analysis into which categories underperform and the potential reasons behind this?

---

> ### Author Response · Authors · 2025-11-27
> **Response to Reviewer Y9Bu**
>
> ## Weaknesses
>
> **Q1:** The framework design lacks detailed explanations. For instance, in the section on the 3DST module: What are the shapes of the intermediate features? Along which dimension are the 3D features of past and current frames concatenated? Why does the spatial encoder take the concatenated 3D features as a whole as input, yet produce spatial features for multiple future frames? Are these features consistent across frames, or is there some kind of autoregressive module involved? Similar questions also arise regarding the 2DST module, and we hope further clarifications can be provided in the subsequent rebuttal.
>
> **A1:** The voxel features have a shape of (B, T, C, H, W, D). To obtain spatial features, we concatenate the past and current frames along the channel dimension (TC), resulting in a tensor of shape (B, TC, H, W, D) which is then fed into the spatial encoder. This spatial feature fusion strategy follows the common practice established in previous works like PreWorld. For temporal encoding, the features are reshaped to (BT, C, H, W, D) before being processed by the temporal encoder. Crucially, our autoregressive LSTM module operates exclusively on these temporal features of shape (BT, C, H, W, D) to generate future temporal features.
>
> **Q2:** The ablation analysis could go deeper. For instance, the 3DST module employs two LSTM modules for temporal forecasting and feature refinement, based on the argument that this "enhances their temporal consistency." However, this design choice is insufficiently validated by the experimental results.
>
> **A2:** We employ two LSTMs in the 3DST module to process the sequence. We did conduct such an experiment during the design phase. When using only the autoregressive LSTM, the average mIoU was 8.51. After incorporating the additional LSTM, the performance was improved to 8.89 mIoU, validating our design choice. We have revised the ablation study section (Table 5) of our paper to more comprehensively validate the effectiveness of our modules.
>
> **Q3:** Despite the abundant set of modular designs incorporated into the model, the performance improvement over previous SOTA methods appears somewhat limited. For example, the average mIoU improvment on Occ3D-nuScenes is only about 3 percent compared to PreWorld+.
>
> **A3:** It is important to note that our baseline model is the PreWorld model with its NeRF rendering branch removed, which yields an average mIoU of only 8.16 (as shown in Table 4). By incrementally incorporating our three key contributions, the performance is substantially improved to 9.83 mIoU. This demonstrates a significant and non-trivial performance gain achieved by our proposed method.
>
> ## Questions
>
> **Q1:** Given the number of components in STM4D, it is notable that its FPS remains comparable to PreWorld. To better understand this efficiency, could the authors provide a detailed breakdown of the latency introduced by each module?
>
> **A1:** Our method achieves comparable FPS to PreWorld while completely eliminating its time-consuming 2D rendering module, demonstrating our approach's superior efficiency. The 3DST module is the most computationally intensive at 88ms (37%), followed by the 2DST module at 62ms (26.1%), while the Lifting and STIM modules are relatively efficient at 47ms (19.7%) and 26ms (10.9%) respectively. The computational time distribution for each module is shown in the table below.
>
> | Module | Time (ms) | Percentage (%) |
> |--------|-----------|----------------|
> | Lifting Module | 47 | 19.7 |
> | 2DST Module | 62 | 26.1 |
> | 3DST Module | 88 | 37.0 |
> | STIM Module | 26 | 10.9 |
> | *Other operations* | *15* | *6.3* |
> | **Total per frame** | **238** | **100.0** |
>
> **Q2:** We note from Table 7 that STM4D does not achieve SOTA prediction performance across all semantic categories. Could the authors provide some analysis into which categories underperform and the potential reasons behind this?
>
> **A2:** For dynamic objects with complex motion patterns such as "motorcycle" (25.15) and "bicycle" (28.49), STM4D's temporal fusion mechanism may be more sensitive to rapid deformations compared to explicit 3D voxel-based methods like TPVFormer, which hold an advantage in modeling the geometric consistency of rigid objects (e.g., "car"). Conversely, STM4D's strong performance on large static classes like "vegetation" (54.18) and "sidewalk" (49.12) highlights its superior semantic segmentation capability. However, this focus might come at the relative cost of fine-grained geometric detail reconstruction for smaller objects, potentially explaining the slightly lower result on categories like "construction vehicle" (22.39) compared to geometry-specialized methods like CTF-Occ (26.93).

---

### Official Review · Reviewer_qeo6 · 2025-11-01

**Soundness:** 3
**Presentation:** 2
**Contribution:** 2
**Rating:** 2
**Confidence:** 4

**Summary:**

This paper introduces STM4D, a unified framework for vision-based 4D occupancy forecasting that addresses the limitation of previous methods by simultaneously modeling spatio-temporal dynamics in both 3D voxel-based representations and 2D multi-view image sequences. The core idea is to leverage the temporal cues inherent in the 2D image sequences to improve the 4D occupancy prediction. The entire architecture is trained end-to-end.

**Strengths:**

1. The technical design is sophisticated and well-justified. The model combines classical techniques like ConvLSTM (extended to 3D) for sequence prediction with modern mechanisms like cross-attention for feature interaction

2. The paper is clearly written, well-structured, and easy to follow.

**Weaknesses:**

1. The paper acknowledges that STM4D underperforms PreWorld+ in terms of the raw IoU metric on Occ3D-nuScenes (Table 1). Furthermore, even with the inclusion of additional 2D labels, the mIoU improvement remains marginal and not sufficiently superior to existing baselines.

2.The proposed 2D/3D spatio-temporal modeling lacks novelty, as similar architectures and strategies have been widely adopted in prior occupancy flow and 4D occupancy forecasting works.

3. While the paper claims that the 2DST module explicitly models spatio-temporal relationships across frames through 3D convolution-based upsampling and refinement, the actual mechanism appears to depend primarily on frame concatenation and a standard 2D CNN (DenseNet). This design choice raises questions about whether the temporal dynamics are being effectively captured, especially compared to the ConvLSTM-based 3DST branch, which is more inherently suited for sequential modeling.

4. There is a lack of comparison and discussion with relevant 4D occupancy forecasting baselines, particularly Let Occ Flow [1]. Since both works share the central idea of leveraging temporal cues from 2D image sequences to enhance 4D occupancy prediction, the claim that this direction “remains unexplored” appears inaccurate or overstated.

5. The architecture introduces two separate LSTM modules to process temporal features, which increases model complexity without a clear ablation demonstrating the necessity or complementary nature of both components.

Reference:
[1] Liu Y, Mou L, Yu X, et al. Let Occ Flow: Self-Supervised 3D Occupancy Flow Prediction. arXiv preprint arXiv:2407.07587, 2024.

**Questions:**

Please see the weakness for details.

---

> ### Author Response · Authors · 2025-11-27
> **Response to Reviewer qeo6**
>
> ### Weaknesses
>
> **Q1:** The paper acknowledges that STM4D underperforms PreWorld+ in terms of the raw IoU metric on Occ3D-nuScenes (Table 1). Furthermore, even with the inclusion of additional 2D labels, the mIoU improvement remains marginal and not sufficiently superior to existing baselines.
>
> **A1:**
> It is important to note that our baseline model is the PreWorld model with its NeRF rendering branch removed, which yields an average mIoU of only 8.16 (as shown in Table 4). By incrementally incorporating our three key contributions, the performance is substantially improved to 9.83 mIoU. This demonstrates a significant and non-trivial performance gain achieved by our proposed method.
>
> **Q2:** The proposed 2D/3D spatio-temporal modeling lacks novelty, as similar architectures and strategies have been widely adopted in prior occupancy flow and 4D occupancy forecasting works.
>
> **A2:**
> Our method, for the first time compared to previous approaches, introduces a multi-view 4D image segmentation task to enhance the performance of 4D occupancy. Unlike methods that rely on other 2D sequences, multi-view image segmentation labels are easier to acquire (e.g., via SAM) compared to other 2D annotations such as BEV segmentation. Moreover, the interaction between our STIM module and the 3D branch helps reduce the dependency on 3D annotations.
>
> **Q3:** While the paper claims that the 2DST module explicitly models spatio-temporal relationships across frames through 3D convolution-based upsampling and refinement, the actual mechanism appears to depend primarily on frame concatenation and a standard 2D CNN (DenseNet). This design choice raises questions about whether the temporal dynamics are being effectively captured, especially compared to the ConvLSTM-based 3DST branch, which is more inherently suited for sequential modeling.
>
> **A3:**
> This is an excellent question. The primary motivation for our 2DST design is computational efficiency, specifically to conserve GPU memory. Furthermore, we empirically explored an alternative architecture for the 2DST that closely mirrors the 3DST module but adapted for 2D features. However, our experiments indicated that this more complex design not only failed to deliver superior performance compared to our straightforward convolutional approach but also introduced a significant increase in memory overhead.
>
> **Q4:** There is a lack of comparison and discussion with relevant 4D occupancy forecasting baselines, particularly Let Occ Flow [1]. Since both works share the central idea of leveraging temporal cues from 2D image sequences to enhance 4D occupancy prediction, the claim that this direction "remains unexplored" appears inaccurate or overstated.
>
> **A4:**
> It is critical to differentiate our work from "Let Occ Flow," which is a model for 3D occupancy estimation, not 4D occupancy forecasting. Our work, to the best of our knowledge, is the first to introduce 2D image sequences as the foundational input for 4D occupancy prediction. A pivotal distinction lies in temporal modeling: their approach utilizes cues from the history of input frames, whereas our method is uniquely designed to capture the evolution of semantic information from the entire sequence, spanning from past to future, which is essential for accurate forecasting.
>
> **Q5:** The architecture introduces two separate LSTM modules to process temporal features, which increases model complexity without a clear ablation demonstrating the necessity or complementary nature of both components.
>
> **A5:**
> That is correct. We employ two LSTMs in the 3DST module to process the sequence. We did conduct such an experiment during the design phase. When using only the autoregressive LSTM, the average mIoU was 8.51. After incorporating the additional LSTM, the performance was improved to 8.89 mIoU, validating our design choice.

---

### Official Review · Reviewer_Uizt · 2025-11-01

**Soundness:** 2
**Presentation:** 3
**Contribution:** 1
**Rating:** 2
**Confidence:** 5

**Summary:**

This paper tackles the task of 4D occupancy forecasting, aiming to predict future 3D scene occupancy states based on past observations. The authors propose a model that performs temporal feature interaction at both 3D and 2D levels, as well as cross-level interaction between 2D and 3D features. Additionally, they introduce an auxiliary task that predicts future-frame 2D segmentation maps to enhance temporal modeling.

**Strengths:**

1. The writing is clear, and the architecture’s structure can be easily understood.

2. The exploration of temporal information in 4D occupancy prediction is meaningful and relevant to current research trends.

**Weaknesses:**

1. The paper does not provide incremental theoretical or conceptual insights to the field. While it applies convolutional operations on 2D and 3D temporal features and uses cross-attention to fuse spatiotemporal information between them, the approach remains largely a straightforward technical combination. There is no clear theoretical analysis or empirical evidence demonstrating the intrinsic advantages or depth of the proposed mechanism.

2. The method performs worse than recent SOTA approaches. For example, II-World [3] achieves 18.97 mIoU on the Occ3D-nuScenes benchmark, roughly twice the performance of this paper’s model. On Cam4DOcc, the improvement over other baselines in the comparison table is only marginal.

3. Although the paper focuses on leveraging temporal information for occupancy forecasting, it should better discuss other recent works that explicitly address temporal fusion in occupancy prediction, such as [1] and [2].

Minor:
Key implementation details such as loss functions and supervision signals should be clearly specified in the Method section to improve understanding.

[1] Rethinking Temporal Fusion with a Unified Gradient Descent View for 3D Semantic Occupancy Prediction, CVPR 2025

[2] GaussianWorld: Gaussian World Model for Streaming 3D Occupancy Prediction, CVPR 2025

[3] II-World: Intra-Inter Tokenization for Efficient Dynamic 4D Scene Forecasting, ICCV 2025

**Questions:**

1. Eqs (3), (4), and (5) appear to be somewhat redundant. In Eq (5), V_{3d} is repeatedly used. From my understanding, could the sum of T_{3d} and T_{3d}' be represented as a residual connection, eliminating the need to explicitly separate the two networks LSTM_{AR} and LSTM_{Refine}? The final residual-connected output could then be concatenated with V_{3d}.

2. In typical occupancy prediction tasks, temporal fusion methods [1, 2] rely on ego-motion information for frame-to-frame spatial alignment. However, the spatial-temporal modeling described in Sec. 3.2 does not appear to use ego-motion cues. Is it challenging to learn spatial-temporal relationships purely from voxel features across frames without motion alignment?

3. L 266–269 mention applying 3D convolution on 2D features. On which additional dimension is this convolution performed, the temporal dimension or another axis? This should be clearly clarified in the paper.

[1] Rethinking Temporal Fusion with a Unified Gradient Descent View for 3D Semantic Occupancy Prediction, CVPR 2025

[2] GaussianWorld: Gaussian World Model for Streaming 3D Occupancy Prediction, CVPR 2025

---

> ### Author Response · Authors · 2025-11-26
> **Response to Reviewer Uizt**
>
> ## Weaknesses
>
> **Q1:** The paper does not provide incremental theoretical or conceptual insights to the field. While it applies convolutional operations on 2D and 3D temporal features and uses cross-attention to fuse spatiotemporal information between them, the approach remains largely a straightforward technical combination. There is no clear theoretical analysis or empirical evidence demonstrating the intrinsic advantages or depth of the proposed mechanism.
>
> **A1:** Our method, for the first time compared to previous approaches, introduces a multi-view 4D image segmentation task to enhance the performance of 4D occupancy. Unlike methods that rely on other 2D sequences, multi-view image segmentation labels are easier to acquire (e.g., via SAM) compared to other 2D annotations such as BEV segmentation. Moreover, the interaction between our STIM module and the 3D branch helps reduce the dependency on 3D annotations. While the cross-attention mechanism in our STIM module is straightforward, our ablation studies (Table 4) confirm its efficacy. We will add a theoretical analysis, including attention visualizations, in the final version to enhance the explanation.
>
> **Q2:** The method performs worse than recent SOTA approaches. For example, II-World achieves 18.97 mIoU on the Occ3D-nuScenes benchmark, roughly twice the performance of this paper's model. On Cam4DOcc, the improvement over other baselines in the comparison table is only marginal.
>
> **A2:** While II-World achieves 18.97 mIoU on the Occ3D-nuScenes benchmark, its approach leverages a pre-trained STCOcc model to first reconstruct past occupancy. Our method operates under a significantly different and more challenging paradigm: it predicts future occupancy end-to-end directly from historical multi-view images, without relying on any 3D occupancy pre-training.
>
> **Q3:** Although the paper focuses on leveraging temporal information for occupancy forecasting, it should better discuss other recent works that explicitly address temporal fusion in occupancy prediction, such as GDFusion and Gaussian World.
>
> **A3:** We appreciate the reviewer's suggestion. Both GDFusion and Gaussian World primarily focus on leveraging temporal information to enhance the voxel features of the current frame. In contrast, our method is dedicated to the temporal fusion of voxel features across multiple frames. While both GDFusion and Gaussian World could potentially be integrated into our framework to generate improved per-frame voxel features, which might lead to performance gains, this integration is not the core innovation of our present work.
>
> ## Questions
>
> **Q1:** Eqs (3), (4), and (5) appear to be somewhat redundant. In Eq (5), $V_{3d}$ is repeatedly used. From my understanding, could the sum of $T_{3d}$ and $T_{3d}'$ be represented as a residual connection, eliminating the need to explicitly separate the two networks $LSTM_{AR}$ and $LSTM_{Refine}$? The final residual-connected output could then be concatenated with $V_{3d}$.
>
> **A1:**
> We thank the reviewer for this suggestion. We have actually explored this approach in our preliminary studies, but our experimental results indicated that it led to a degradation in performance.As presented in the following table:
>
> | Metric  | Method | 1s     | 2s     | 3s     |
> |---------|--------|--------|--------|--------|
> | **IoU** | Preliminary| 23.14  | 20.74  | 19.25  |
> |         | Now | 23.46  | 21.04  | 19.51  |
> | **mIoU**| Preliminary | 12.18  | 9.17   | 7.46   |
> |         | Now | 12.48  | 9.41   | 7.60   |
>
> **Q2:** In typical occupancy prediction tasks, temporal fusion methods [1, 2] rely on ego-motion information for frame-to-frame spatial alignment. However, the spatial-temporal modeling described in Sec. 3.2 does not appear to use ego-motion cues. Is it challenging to learn spatial-temporal relationships purely from voxel features across frames without motion alignment?
>
> **A2:**
> In the lifting module, we employed ego-motion information to align/warp the voxel features from previous timestamps to the current frame's coordinate system. We acknowledge the lack of clarity in the manuscript regarding this step and will revise the description accordingly in the final version.
>
> **Q3:** L 266–269 mention applying 3D convolution on 2D features. On which additional dimension is this convolution performed, the temporal dimension or another axis? This should be clearly clarified in the paper.
>
> **A3:**
> We apply 3D convolution to the 2D features, where the additional dimension refers to the temporal dimension. We will revise the description in our paper to make this clear.

---

### Note · Authors · 2026-03-28

I have read and agree with the venue's withdrawal policy on behalf of myself and my co-authors.

---

### Meta-Review · Area_Chair_jqfD · 2025-12-16

**Summary:**

This paper introduces STM4D, a unified framework for vision-based 4D occupancy forecasting that addresses the limitation of previous methods by simultaneously modeling spatio-temporal dynamics in both 3D voxel-based representations and 2D multi-view image sequences. The core idea is to leverage the temporal cues inherent in the 2D image sequences to improve the 4D occupancy prediction. The entire architecture is trained end-to-end.


All the reviewers tend to be negative. They pointed out that the proposed method lacks theoretical or conceptual insights (Uizt, qeo6, KpCi), results are worse than recent SOTA methods (Uizt, qeo6, Y9Bu), more related references should be discussed or compared  (Uizt, qeo6, KpCi).

The rebuttal did not address those key concerns. It is suggested to justify the novelty by discussing previous similar works in detail and highlight novel insights the proposed method and experiment can provide. It is also benefitial to train the model with the same recipe, which may help improve the comparison with sota methods.

**Reviewer Concerns:**

The concerns regarding novelty and performance are still outstanding.

Reviewers raise concerns that 'the approach remains largely a straightforward technical combination‘, 'similar architectures and strategies have been widely adopted in prior occupancy flow and 4D occupancy forecasting works', ' similar insights have already been explored and verified in prior occupancy-related literature.'


The comparison with EfficientOCF (CVPR 2025) is solved during the rebuttal, however, reviewers' concern on the worse performance than II-World (ICCV 2025)  still hold.

**Reviewer Scores:**

I think reviewers may choose to keep their negative score (2244) as their key concerns on novelty still remain and are difficult to be solved.

---

### Decision · Program_Chairs · 2026-01-26

Reject